# Possibilities of an Electronic Nose on Piezoelectric Sensors with Polycomposite Coatings to Investigate the Microbiological Indicators of Milk

**DOI:** 10.3390/s24113634

**Published:** 2024-06-04

**Authors:** Anastasiia Shuba, Ruslan Umarkhanov, Ekaterina Bogdanova, Ekaterina Anokhina, Inna Burakova

**Affiliations:** 1Department of Physical and Analytical Chemistry, Voronezh State University of Engineering Technologies, Revolution Avenue 19, 394000 Voronezh, Russia; rumarkhanov@gmail.com; 2Department of Technology of Animal Products, Voronezh State University of Engineering Technologies, Revolution Avenue 19, 394036 Voronezh, Russia; ek-v-b@yandex.ru; 3Laboratory of Metagenomics and Food Biotechnology, Voronezh State University of Engineering Technologies, 394036 Voronezh, Russia; katya_anoh@mail.ru (E.A.); vitkalovai@inbox.ru (I.B.)

**Keywords:** piezoelectric sensor, composite coating, volatile organic compounds, milk, microbiological indicators, chemometrics, regression

## Abstract

Milk and dairy products are included in the list of the Food Security Doctrine and are of paramount importance in the diet of the human population. At the same time, the presence of many macro- and microcomponents in milk, as available sources of carbon and energy, as well as the high activity of water, cause the rapid development of native and pathogen microorganisms in it. The goal of the work was to assess the possibility of using an array of gas chemical sensors based on piezoquartz microbalances with polycomposite coatings to assess the microbiological indicators of milk quality and to compare the microflora of milk samples. Piezosensors with polycomposite coatings with high sensitivity to volatile compounds were obtained. The gas phase of raw milk was analyzed using the sensors; in parallel, the physicochemical and microbiological parameters were determined for these samples, and species identification of the microorganisms was carried out for the isolated microorganisms in milk. The most informative output data of the sensor array for the assessment of microbiological indicators were established. Regression models were constructed to predict the quantity of microorganisms in milk samples based on the informative sensors’ data with an error of no more than 17%. The limit of determination of QMAFAnM in milk was 243 ± 174 CFU/cm^3^. Ways to improve the accuracy and specificity of the determination of microorganisms in milk samples were proposed.

## 1. Introduction

Among the huge number of different products of animal and plant origin, the most valuable from a nutritional and biological point of view are milk and dairy products. They are able to meet the body’s needs for calcium, phosphorus and riboflavin. In addition, they contain complete proteins, the high biological value of which is due to the composition, the balance of essential amino acids, good digestibility and assimilation (96–98%). A comparison of the composition of the essential amino acids of milk proteins with the composition of the “ideal” protein indicated that they practically do not have amino acids that limit their biological value. In addition, milk contains many enzymes, water-soluble vitamins, minerals and other important nutrients necessary for normal metabolism [1]. However, the most common problem associated with the consumption of dairy products by various age groups of the population is their intolerance, which is caused by a wide variety of disorders in the cleavage and absorption of carbohydrates, fats or proteins. Moreover, hypersensitivity reactions to milk are mainly caused by protein components and are primarily due to the presence of whey proteins in milk [2,3,4]. Fats and lactose are not characterized by antigenic properties.

At the same time, milk is a good nutrient medium for most microorganisms, both those introduced via starter cultures and those that come from outside. Therefore, modern regulatory documents place high demands on the quality of milk and dairy products [5,6,7]. The levels of microorganisms in dairy products intended for release into circulation on the territory of the Customs Union should not exceed the established requirements [7]. In the case of dairy products, special attention is paid to the control of QMAFAnM, yeasts and molds, as well as the presence of pathogenic microorganisms, including *Salmonella*, *Staphylococci* and *Listeria*. This is due to their ability not only to adapt and reproduce in the host’s body but also their ability to release pathogenic determinants into the medium, as well as producing toxins that cause food allergies and intoxication [8]. In addition, these substances are quite stable and can retain their allergenic properties even after processing [9].

When conducting state supervision over compliance with mandatory requirements, microbiological tests are the most problematic, since standard methods of microbiological research are quite laborious and take several days to obtain a result [10,11,12,13]. Therefore, the development and practical implementation of express methods for qualitative and quantitative assessment of the main microbiological safety indicators, including potential sources of allergens, is vital task for today.

In recent years, methods for the rapid detection and monitoring of pathogenic microorganisms in milk and dairy products have been actively developed. Along with direct methods based on determining the composition of intracellular compounds and the cell wall components of microorganisms (for example, polymerase chain reaction [14], loop isothermal amplification of nucleic acids [15,16] and bioluminescence of the resulting adenosine triphosphate [17]), special attention has been paid to indirect methods of analysis based on recognition of the metabolic products of various microorganisms using sensors [18,19], including antibodies [20]. Moreover, a system of piezoelectric microelectrode arrays modified with conductive polymers has also been developed to monitor the bacterial contamination of fresh milk in real time in the concentration range of 10^3^–10^6^ CFU/mL; the detection limit is 10^2^ CFU/mL [21].

In the literature, there are some works on the use of sensor arrays to determine the number of pathogens in food products [22,23,24] and milk [24,25], as well as the emitted toxins [26]. Electronic noses are widely used for predicting shelf life, detecting microbial spoilage of milk, classifying milk by brand and type, determining the fat content and monitoring the quality of milk by detecting off-flavors [27,28]. Today, studies into the possibility of using an electronic nose to detect pathogenic microorganisms in milk are few and far between and are at an early stage of development. The main source of milk contamination is the presence of mastitis in cows, with volatile compounds released by the main pathogens that cause mastitis [29] and contaminated milk [30] being indicative; these can be used as markers in the assessment of microbiological safety [31]. When using sensors to quantify microorganisms in milk, not only the relationship between relative signals of the sensor and the growth in the number of bacteria in the sample under various storage conditions (refrigerator, room temperature) and technological regimes should be accounted for [28,31,32], but also the dependence of the composition of the milk’s volatile compounds on the physicochemical parameters and seasonality [33]. Works on the use of sensor arrays for assessing the quality of milk and dairy products are known [34,35,36], including the total bacterial contamination of milk [25,28,32]. With a fairly large number of works, the introduction of such methods is limited, due to insufficient knowledge of the mechanisms of the formation of volatile compounds and their connection with the milk’s microflora. In some works [32,34,35], pathogenic microorganisms were determined in samples of pasteurized milk or artificially created standards with a given composition, in which the presence of other microorganisms was minimal; therefore, the results thus obtained cannot be applied to the analysis of raw milk. Consequently, at the moment, the relationship between the gas composition of raw milk and sensor signals of microbiological indicators has not been established, and the possibility of the influence of the species composition of the microflora in milk on the output data of the sensors, which is significant for assessing bacterial contamination, has not been assessed.

Among the frequently used sensors in food analysis are piezoelectric quartz sensors. They can be used for the analysis of liquid and gaseous media. Their operating principle is based on the attenuation of a volumetric acoustic wave during the sorption of components on the surface of an electrode glued to an AT-cut quartz crystal. The amount of attenuation is proportional to the mass of the substance sorbed on the surface of the crystal. The selectivity of the sorption of substances is regulated by the nature of the film deposited on the resonator’s electrodes [37,38,39].

When analyzing mixtures with a complex composition, for example, food products, it is important to select the coatings that selectively sorb the target components. Thus, when determining microorganisms in food products, volatile metabolites produced by microorganisms and changes in the ratio of volatile substances in the product matrix, which are associated with the destruction of the macrocomponents of the product, can be detected. Even though the released volatile metabolites are not strictly specific for many microorganisms, it is the complex of changes associated with the concentrations and ratio of volatile components in the gas phase that presumably depends on the type of microorganism and the composition of the product. It can allow us to identify the presence and quantity of microorganisms in products that exceed the standard values.

The goal of this work was to evaluate the possibilities of using an array of gas sensors with polycomposite coatings to assess the microbiological indicators of milk safety and to compare the microflora of milk samples.

## 2. Materials and Methods

This section describes the method of forming sensor coatings, the methods for analyzing milk and the methods used for mathematical processing of the results obtained.

### 2.1. Objects of the Research

Various classes of volatile organic compounds were selected as the test substances for assessing the characteristics of the sorption coatings of sensors: alcohols (ethanol, butanol, isobutanol and isopentanol), ketones (acetone and butanone-2), ethyl acetate, acetaldehyde, carboxylic acids (formic, acetic and butyric) (analytical grade, Reachim LLC, Moscow, Russia) and distilled water, as well as aqueous solutions or emulsions of some volatile compounds in the concentration range of 0.001–10% by volume. Moreover, the objects of the research were samples of raw cow’s milk (n = 14) obtained from five farms in different seasons, cooled immediately after milking to T = (4 ± 2) °C and delivered to the laboratory for no more than 3 h of storage, and samples of the standards (artificial milk samples with a known and constant composition). The standards were prepared by sequentially dissolving the components (cream with a fat mass fraction of 20%, JSC Avida, Russia; milk protein concentrate, PJSC MK Voronezh Kalacheevsky Cheese Plant, Russia; anhydrous lactose, JSC Lenreaktiv, Russia; calcium lactate 5-water, JSC “Vekton”, Russia; calcium dihydrogen phosphate 1-water, JSC “Vekton”, Russia; skim milk powder, PJSC MK Voronezh Kalacheevsky cheese plant, Russia) in distilled water. The composition of Standard No. 1, based on the individual components was as follows: mass fraction of fat, 4.0–4.1%; mass fraction of protein, 3.0–3.1%; mass fraction of lactose, 4.65–4.70%; mass fraction of ash (in the form of a mixture of calcium lactate and calcium dihydrogen phosphate), 0.60–0.65%; mass fraction of dry substances, 12.3–12.5%. The composition of Standard No. 2, based on skimmed milk powder and cream, was as follows: mass fraction of fat, 4.05–4.10%; mass fraction of protein, 3.40–3.45%; mass fraction of lactose, 4.75–5.30%.

### 2.2. Analysis of the Gas Phase by the Sensor Array

#### 2.2.1. Instruments and Measurement Methods

The study of the gas phase of milk samples was carried out on two devices based on piezoelectric quartz sensors with different ways of inputting the gas phase. These were the MAG-8 (LLC Sensors–New Technologies, Voronezh, Russia) with injected input of the gas phase and the Diagnost-Bio-8 (LLC Sensino, Kursk, Russia) with frontal input of the gas phase. For injected input, the volume of the equilibrium gas phase over the samples was 3 cm^3^ and the time of measurement was 80 s, for frontal input, the volume of the samples was 0.05 cm^3^ and time of measurement was 120 s (40 s was for sorption, 80 s was for desorption). Moreover, to reduce the time drift of the sensors on the days of analyzing the milk samples, samples of double-distilled water, as the main component of milk, were also measured. Gas phase measurements of each sample, including the standards, were carried out three times. The total number of measurements of raw milk samples was 48. Additionally, to increase the emission of volatile compounds from a milk sample, we also measured the gas phase of milk after treating it with ultrasound (processing time, 180 s; power, 50 W). Measurements of the gas phases of sonicated milk samples were also carried out three times using the MAG-8 sensor.

#### 2.2.2. Formation of the Sensor Coating

The surface of the electrodes of a quartz resonator with a base frequency of 14.0 MHz (Meteor Plant JSC, Volzhsky, Russia) [39,40] were coated with composite films of several different natures, previously proposed for the analysis of the gas phase of milk [41,42]. The solutions of the sorbents in suitable solvents were prepared with a concentration of 5 mg/mL and mixed in a proportion of 1:1 by volume. Solvents of the intrinsic phases (toluene and ethanol) were used according to the classifications of analytical grade (ReaChem, Russia). All chromatographic phases were purchased from the company Alfa Aesar, Haverhill, USA; chitosan (pH = 5.2, sodium nitrate, Mr = 3000 kDa) was obtained from the laboratory of Prof. V.P. Varlamov, Federal Research Center Fundamentals of Biotechnology of the Russian Academy of Sciences; the micellar casein concentrate (CMC) was obtained from Molvest JSC. These compounds were chosen as the sorbents due to their high affinity for milk components. Thus, chitosan is often used to create biocompatible films that are safe for products, including those commonly distributed in milk [43], and it has a high sensitivity to ketones and aldehydes [44,45], which are present in the gas phase of milk [46]. Chromatographic phases were also selected, taking the use of these sorbents in the analysis of aroma-forming components of milk into account [47]. Deep eutectic solvents were also used to determine the residues of antibiotics and hormones in milk and dairy products [48,49]. Consequently, these compounds interact well with milk components and can be effective when interacting with the components of the gas phase of milk.

The quartz resonator’s electrodes were preliminarily degreased with a solvent (acetone or chloroform) and dried in a drying oven. After cooling to room temperature in the desiccator, the electrodes were coated. The coatings were formed by dispersion spraying from solutions of sorbent mixtures [50]. A mixture of sorbent solutions was placed in a container with a conical protrusion at the bottom; a conical filter with a hole at the top of 10 μm and a divider was put on top (Figure 1). The dried laboratory air was supplied under a pressure of 120 kPa to a container with a mixture of the sorbents. The resulting aerosol was sprayed on the electrodes of the piezoquartz resonator for 5 s on each side, after which, they were placed in a drying oven at a temperature of 50 °C for 1 h until the unbound solvent was completely removed. Deep eutectic solvents and the films based on them were formed according to the method described earlier [41]. After the sensors had cooled, their frequency of oscillation was measured and the masses of the resulting coatings were calculated according to the Sauerbray equation [40]. The characteristics of the resulting sensors are presented in Table 1.

Sensor noise was assessed as the shift in the vibration frequency of the quartz resonator for 80 s without sample loading, and drift was assessed as a shift in the vibration frequency after 6 months of active operation. Estimation of the effectiveness of the sorption by the composite coatings was assessed using specific mass sensitivity and the selectivity coefficient [50]. The surface of the obtained coatings was examined via the Solver-Pro NT-MDT, Moscow, Russiascanning tunneling microscope.

#### 2.2.3. Processing the Sensors’ Output Data 

In the standard software, the output curves of the sensors were recorded in the form of chronofrequencies, which were analyzed using built-in data processing algorithms as maximal changes in the vibration frequency during the measurement (−ΔF_max,i_, Hz) (Figure 2).

The analytical signals obtained from the sensors and the signals of the sensors at different moments of sorption were used to calculate the additional sorption parameters A_ij∑_ and β. The principles of calculating the additional parameters were described in more detail in other works [41,51,52]. The calculated parameter A_i/j_, as shown earlier [53], mainly characterizes the qualitative composition of the gas mixture. It was shown that the parameters A_ij∑_ and β_i_ can be used as identification parameters, subject to certain conditions and assumptions. The main assumptions and conditions include strict adherence to the methods of forming coatings on the electrodes of piezoelectric resonators, ensuring the linearity of the sensors’ response (the constancy of the sensitivity of microweighing) in the selected range of concentrations of the detected substance. For each parameter A_ij∑_, the coincidence criterion was calculated to identify the substances. The calculated values of the parameter A_i/j_ and the coincidence criterion were determined for the given experimental conditions, accounting for the nature of the volatile substances, the composition of the sample, and the mode of input of the gas phase to the detection cell. An earlier study proved that the calculated parameter A_ij∑_ of sensors with composite coatings can also be used for the identification of substances [51]. In total, 69 additional parameters were calculated, accounting for the features of the kinetics of the sorption of the volatile compounds on composite coatings, reflecting the interaction of each sorbent in the coating with a volatile compound. The list of formulae for calculation of the parameters is given in Table A1.

### 2.3. Determination of the Physical and Chemical Properties of the Milk

The mass fraction of dry solids in the samples was determined by drying [5] in a Binder ED 53 oven (BINDER Inc., Tuttlingen, Germany) to a constant mass at T = (105 ± 2) °C; the mass fraction of fat was determined by the Gerber acid method [5], the mass fraction of total protein was determined by titration of formol [6], density was determined by the areometric method [5], titratable acidity was determined by the titrimetric method with a phenolphthalein indicator [5], the purity group was determined by the gravimetric method [5], the size of the milk fat globules was determined by 0020 microscopy (the microscope was an Altami Bio 1, Altami Ltd., Saint Petersburg, Russia, with a Canon camera adapter, Canon Inc., Tokyo, Japan) at a magnification of ×1200 using Gorjaev’s count chamber. All chemicals were of analytical grade (Stock Company Lenreactiv, Saint-Petersburg, Russia). The experimental studies of each sample were carried out 3–5 times. The number of repetitions of each experiment to determine one value was three times. Calculations were carried out by mathematical statistical methods using the XLSTAT application (Lumivero, Denver, CO, USA) of the Microsoft Office 365 Family (Microsoft Corporation, Redmond, WA, USA). Data were expressed as the mean ± standard deviation for normally distributed data. A *p*-value less than or equal to 0.05 was used to determine if the findings were significant.

### 2.4. Determination of Microbiological Indicators

Microbiological indicators (the quantity of mesophilic aerobic and facultative anaerobic microorganisms (QMAFAnM) and the quantity of yeasts and molds) were determined using microbiological inoculation on universal nutrient media (plate count agar, Sabouraud agar, Obolensk, Russia) according to the standard methods describing in GOST [7,54,55]. QMAFAnM was estimated using three dilutions of milk (from 10^6^ to 10^4^). The raw milk sample was diluted 10 times to estimate the quantity of yeasts and molds. To identify the species diversity of the milk microflora for each farm, individual colonies of microorganisms were additionally grown on universal nutrient media at a temperature of 37 °C for 3 days. The gas phase of the microorganisms grown on Petri dishes after counting the colonies was also measured using an array of sensors with the mode of frontal diffusion into the detection cell.

### 2.5. Sanger Sequencing and Bioinformatic Analysis

Molecular genetic studies were also carried out to determine the possible presence of opportunistic bacteria, namely enterohemorrhagic *E. coli* (*EHEC*), *Salmonella* spp. and *Listeria monocytogenes.* In the first stage, total deoxyribonucleic acid (DNA) was isolated from the obtained samples using the Proba-GS commercial kit (DNK-Technology, Moscow, Russia) according to the manufacturer’s protocol. The concentration was then measured for each sample using a Qubit fluorimeter (Thermo Fisher Scientific, Bothell, WA, USA) and a commercial Qubit™ dsDNA Quantification Assay Kit (Thermo Fisher Scientific, USA). Detection of enterohemorrhagic *E. coli* (*EHEC*), *Salmonella* spp. and *Listeria monocytogene* was carried out using commercial reagent kits for detecting the DNA of these bacteria using the polymerase chain reaction (PCR) method. The components of the reaction mixture and the amplification conditions were chosen according to the manufacturer’s protocol.

After microbiological inoculation, pure cultures, including both fungal and bacterial cultures, were obtained to identify the species of the grown colonies. DNA was extracted from these cultures using the commercial Proba-GS kit according to the manufacturer’s instructions. After that, amplification was carried out for each bacterial sample using a commercial mix of 5X ScreenMix-HS (Evrogen, Moscow, Russia) and a universal bacterial primer. For each mold sample, amplification was carried out with a universal mold primer. The sequences of the primers are presented in Table 2.

The conditions for the amplification of these sites were as follows: initial denaturation at 94 °C for 4 min, then 39 cycles of total denaturation at 94 °C for 20 s, annealing of the primers at 54 °C for 30 s and elongation at 72 °C for 50 s. Subsequent visualization of the obtained amplification products was carried out using the method of gel electrophoresis in a 2% agarose gel in a 1× Tris-acetate buffer. Subsequently, the amplification products were mechanically cut out of the agarose gel for subsequent purification, which was carried out using the commercial Cleanup Standard kit (Evrogen, Russia) according to the manufacturer’s protocol. Then sequence amplification was carried out with the purified PCR products using the ITS4 (5′ TCC TCC GCT TAT TGA TAT GC 3′) and 1100R (5′ GGG TTG CGC TCG TTG 3′) primers and a reaction mixture of the Quantum Dye Terminator Cycle Sequencing Kit v3.1 (Thermo Fisher Scientific, WA, USA) according to the manufacturer’s instructions.

The amplification protocol was as follows: initial denaturation at 96 °C for 1 min, then 25 cycles of total denaturation at 96 °C for 10 s, annealing of the primers at 50 °C for 5 s and elongation at 60 °C for 4 min.

After that, the amplification products were purified using the commercial reagent BigDye XTerminator™ Purification Kit (Thermo Scientific, USA) according to the manufacturer’s protocol.

Sequencing was carried out using the NANOFOR 05 genetic analyzer (Sinthol, Moscow, Russia). As a result of sequencing, chromatograms were obtained for all samples, the sequences of which were checked using the Chromas program. The results were then analyzed using the Basic Local Alignment Search Tool (BLAST) (https://blast.ncbi.nlm.nih.gov/Blast.cgi: (accessed on 26 July 2023)) to find similarities between the sequences and those in the databases.

### 2.6. Mathematical Processing of the Experimental Results

The limit for the determination of volatile compounds (LOD) in aqueous solutions was estimated using regression equations of the dependence of the sensors’ signals on the concentration of volatile compounds. The value of the concentration of the volatile compound, for which the predicted value of the sensor’s signal was 3.3σ times greater than in water vapor, was considered the limit of determination. To compare the data obtained from an array of sensors under different sorption modes and to identify volatile substances in the gas phase above the samples, we used the similarity parameter δ, which was calculated according to a previously proposed approach [56].

To reduce the effect of time drift on the sensor’s signals, the sensor’s outputs normalized to the average analytical sensor’s signals for water vapor were used for mathematical data processing. The signal of the sensor obtained during measurement of the milk sample was divided to average the sensor’s signal for water vapor obtained on the day of analysis. The data matrix based on the results of the analysis of milk samples for subsequent processing by multivariate analysis methods was formed from the following outputs of the sensors: full chronofrequency sensors for measurements in the injected gas phase mode, analytical sensor signals (−ΔF_max,i_) and the calculated parameters A_ij∑_ and β_i_ selected from the results of the analysis of the vapors of volatile compounds and their aqueous solutions. In total, the initial sensor data matrix turned out to be 701 × 48 for each measurement method (frontal and gas phase injection). As was shown earlier, the inclusion of calculated parameters in the original data matrix for its processing by projection methods allows us to reduce the prediction error due to more accurate consideration of the composition of the gas phase, especially for the classification problem [53,56]. Next, the number of variables was optimized in the model by excluding those with low values for the regression coefficients and high leverage values. For each model, the process of optimizing the number of variables was carried out completely.

The values of the decimal logarithm of QMAFAnM (quantity of yeasts or molds) were used as the predicted indicator. The data matrix was processed using the module for Microsoft Excel (Chemometrix-Add-in) and Unscrambler version 10.0.1 (CamoSoftware AS, Oslo, Norway). To reduce noise, the output curves of the sensors were preprocessed by the Savitsky–Golay filter using a three-point approximation and normalized to the maximum signal of the sensor in water vapor measured on that day. Before using multivariate regression methods, the sensor’s data were subject to block scaling (all data, except for the calculated parameters, were automatically scaled by the standard deviation, and centering on the mean value was used only for the calculated parameters).

To build the regression models, the method of projection on the latent structures was chosen. The method of full cross-validation was chosen as an algorithm for checking the models.

## 3. Results

### 3.1. Characteristics of the Sensors

Among the main characteristics of sensor coatings that determine the sorption properties of a sensor are its porosity and surface characteristics, as they determine the magnitude of nonspecific interactions with the adsorptive material. The porosity and surface characteristics influence the number of centers of physical sorption, since they characterize, in the first approximation, the value of uncompensated surface energy. Since it was previously shown that during intensive use, the greatest changes were observed for sensors with a high affinity for the sample’s components, films of the most hydrophilic coatings were studied using atomic force microscopy during the measurement period.

The results of the examination of the most hydrophilic sensor’s surface using atomic force and scanning electron microscopes are presented in Table 3 and Figure 3. It has been shown that the average roughness and other morphological characteristics of the surface increased after training the sensors in vapors of the test substances and during active use (Table 3). It was also found that all the morphological characteristics of the choline + sorbitol coating changed more significantly during operation.

Based on the results of testing in the vapors of pure volatile organic compounds and the equilibrium gas phase over their aqueous solutions, the detection limits and other sorption characteristics of the sensors were determined (Table 4 and Appendix A).

It was also found that the coefficient of variation of the relative signals of the sensors with coatings based on deep eutectic solvents in the first few days after manufacturing the sensors reached 58%; after 14 days of training in the vapors of pure volatile compounds, the coefficient of variation of the normalized signals of the sensors did not exceed 5%. The normalized signals of the sensors in water vapor for each measurement relative to the average signal of the sensor after 14 days of operation was 1.05–0.95 (Figure A1).

Coatings were characterized by low limits for the determination of volatile compounds in aqueous solutions and rather high specific selectivity coefficients for the carboxylic acids. The detection limits of substances corresponded to the range of concentrations of these compounds in milk [57,58,59,60,61]. The lowest detection limits for volatile substances from other studied sensors were for sensors coated with erythritol, as was established (Table 4).

Analytical information from the sensor array was also compared, depending on the method of introducing the gas phase into the detection cell. When the gas phase is examined for raw milk samples via the frontal method, the sensor’s signals were 3–6 times higher than during the injection of the gas phase, but the calculated parameter of the efficiency of sorption Aij [41,53] and the coefficient β_i_ [41] were similar (Figure 4).

The similarity parameter δ for the calculated sorption parameters was 80% and 94.2%, which is satisfactory, considering the coefficients of variation of the sensor’s signals under the different measurement conditions. Consequently, it was possible to consider the established patterns of sorption of the volatile compounds under the different conditions of inputting the sample to be identical, and we compared the calculated parameters for different measurement conditions with each other.

### 3.2. Microbiological Indicators of Milk

It was established that the samples conformed with requirements of the regulatory documents for raw milk in the Russian Federation [5,6,7], except for samples Nos. 7, 9 and 11–13. These samples had a mass fraction of total protein less than minimum of 2.8% and a titratable acidity less than the standardized 16 ⁰T. All samples were in the first class of purity. No opportunistic bacteria were found in the milk samples. The values of all the estimated physical and chemical properties of the raw milk samples, including the size of fat globules, are presented in the Appendix A. The results of determining the microbiological parameters of the milk samples from the farms are presented in Table 5.

Microorganisms isolated in the microflora of raw milk were also identified using the molecular genetic method.

The results of determining the species diversity of microorganisms in the milk microflora from the farms are presented in Table 6.

Sporulation of the microorganisms can also influence the amount of volatile compounds released by the microorganisms and their ratio in the gas phase.

Based on the calculated parameters of the sensor array, profiles were constructed for each farm (Figure 5). Changes in the values of the calculated parameters for the sensors varied within 5–20%, depending on the parameters’ value; the larger the parameters’ value, the higher the coefficient of variation.

It was shown that the composition of the gas phase of the microflora of milk from farm No. 4 differed markedly from the gas composition of all other samples; the gas composition of the microflora of milk from farms No. 1 and No. 5 were similar to each other. The gas composition of the microflora of milk from farm No. 3 differed from the others, while the parameters’ values were small. The gas composition of the microflora of milk from farm No. 2 also differed from the rest and was characterized by the maximum values of the parameters β_6_ and β_7_. When comparing the profiles of the calculated parameters for microorganisms with the profiles of the calculated parameters for milk samples from these farms, the samples also differed from each other and differed from the profiles for microorganisms. The profiles for the samples from farms No. 3 and No. 5 were as close as possible in terms of the similarity parameter (65%).

Since it was previously shown that the sensor’s signals are influenced by the composition of the macrocomponents of milk [42], to predict the microbiological parameters of milk, along with signals from the output curves of the sensors, the calculated parameters were used. The characteristics of the resulting models are presented in Table 7.

For QMAFAnM, the greatest linearity was achieved, and thus the smallest error, while the number of variables used was the largest. The detection limit of QMAFAnM according to the model in terms of CFU was 243 ± 174 CFU/cm^3^. It was previously shown that when milk samples are processed via ultrasound, the sensor’s signals correlate with the content of mold and fungi in milk [42]. Therefore, to predict the content of yeast and mold, a matrix of the sensor’s data was used, based on the results of the measurements of milk samples after ultrasonic treatment.

The regression coefficients for each model are presented in the Appendix A.

These models were tested on four samples of raw milk from the same farms, for which the microbiological indicators were also determined and the gas phase was measured by an array of sensors, but which were not included in the training set. The results of prediction are presented in Table 8.

Considering the calculated values of deviation according to the model, the predicted values of the microbiological indicators coincided with the values determined by the standard methods.

For all milk samples, the profiles of the calculated parameters of the sensor array were also constructed, which were the most informative in the regression models (Figure A2). In the diagram, two areas can be roughly distinguished: one of them is characterized by small changes between samples, mainly parameters calculated from the signals of two sensors; the other is characterized by significant differences between samples in the parameters calculated from the signals of one sensor. It was shown that the strongest deviation between samples was typical for sensors No. 3 and No. 6. This may have been due to the high affinity of PVP and CMC for water vapor and compounds associated with the protein and fat content in the milk samples [42]. The parameters of sensor No. 4 were most significant when used in calculations with the parameters of other sensors (No. 5, No. 7 and No. 3). This was due to the fact that sensor No. 4 was the most hydrophilic of the sensors studied. The ratio of its parameters to the parameters of less hydrophilic sensors made it possible to estimate the relative content of various classes of substances in the gas phase above the sample, namely alcohols and aldehydes (sensor No. 7), fats (sensor No. 3) and acids (sensor No. 5).

## 4. Discussion

It has been established that coatings based on DESs (deep eutectic solvents), together with amorphous silicon oxide, have a more developed surface, the DESs is uniformly distributed over the surface and the average roughness increases slightly. In the first two weeks of training the coatings in the vapors of pure volatile compounds and water, similar to coatings with polymer films, the surface of the coatings based on DES changed along with the sorption properties. After 2 weeks, the reproducibility of the sensor’s signals reached 5% and remained virtually unchanged throughout the entire period of operation; therefore all structural changes to the surface had been completed. According to the AFM and SEM data (Figure 2, Table 3), it was established that after 6 months of operation, the roughness of the coating containing the crystals of choline + sorbitol increased due to the increase in the peaks and the smoothing out of small areas of roughness, including those due to an increase in the size of DES particles (these swelled when we analyzed the gas phase of aqueous solutions).

Even though the selectivity coefficients for isopentanol and butanone-2 vapors, determined from pure substances, were low, their limits of determination were also quite low (Table 4), which was presumably because in aqueous solutions, they were concentrated at the interface and, together with the associated water, were sorbed on the coatings. This effect was observed when studying the sorption of the equilibrium gas phase of highly dilute solutions of volatile substances (less than 0.01% vol.). The compounds selected to determine the detection limits were the most widely known volatile markers associated with the presence of pathogenic microorganisms in milk [62,63,64,65,66,67]. In relation to the other volatile compounds, the array of sensors was characterized by a high specific mass sensitivity of microweighing (Appendix A); the selected compounds were also present in the gas phase of milk, as in the natural changes associated with the proliferation of microorganisms during storage [61,62] and with the changes associated with the technological process of milk processing [68,69,70].

Since it was more convenient to use different versions of the measuring device for different objects of analysis, the hypothesis of the convergence of the measurement results with frontal and injected input of the gas phase into the detection cell was confirmed, taking changes in the desorption rates of substances in open and closed detection cells into account (Figure 3). Thus, the milk microorganisms grown on nutrient media were analyzed with the frontal mode of input of the gas phase into the detection cell, and the raw milk samples were analyzed with an injected gas phase.

Certain microbiological indicators of milk samples from one farm varied (Table 5) and exceeded the number of microorganisms in raw milk allowed by the regulatory documents; however, the species diversity of microorganisms grown on nutrient media did not change for each farm (Table 6). The physicochemical parameters of the milk also varied within the same farm (Appendix A), which may have affected the composition of the gas phase, especially the number and size of fat globules in the milk [70]. Therefore, for constructing the profiles, the calculated parameters were selected if they did not correlate or were less related to the physicochemical parameters of milk and related to a greater extent with microbiological parameters (Figure 4). It was found that the profile of the calculated parameters for the sample from farm No. 3 (Figure 4a) was the most different from the others (minimum parameter values), which may have been due to the presence of bacteria of the species *Bacillus thuringiensis* in the milk sample (Table 6), which, according to data from the literature, are capable of suppressing the growth of food pathogens, as well as the bacteria that cause mastitis in cattle [71], which led to a decrease in the number and diversity of volatile compounds in the gas phase of the milk and, as a consequence, a decrease in the values of the calculated parameters. The profiles of the calculated parameters of the sensor array for raw milk samples differed significantly from the profiles of the calculated parameters for microorganisms (Figure 4a,b), which was apparently due to the presence of yeast and fungi in the milk samples, as well as the influence of native components forming the smell of milk, against which it is currently impossible to distinguish a profile for specific microorganisms. To increase the specificity of the profiles of the calculated parameters for specific microorganisms, a more detailed study of their relationship with the composition of the components of milk, including amino acids and carbohydrates, is necessary, as the main sources of the volatile compounds in the metabolism of microorganisms. Using such profiles, it would be possible to create metabolic atlases for bacteria in food products, similar to the creation of atlases for identifying diseases and pathologies [72,73]. In this case, the total content of microorganisms was predicted well using the proposed parameters and output curves of the sensors (Table 7 and Table 8).

The study of the profiles of the calculated parameters of the array of sensors (after optimization when constructing the regression) for all milk samples indicated that in order to predict the microbiological parameters, it is necessary to consider the sensor’s signals that correlate with the protein and fat content (No. 2 and No. 3 [42]), as well as the parameters reflecting the composition of the microflora of milk. In this case, it is possible to exclude the parameters and signals of sensors No. 1 and No. 8 for predicting the microbiological parameters, with a slight increase in the prediction error of 2–3%. Thus, by using chemometrics methods, it is possible not only to determine food falsification [74] but also food safety indicators. The use of other chemometrics algorithms [75,76] could be useful to reduce the prediction error.

## 5. Conclusions

The work demonstrated the possibility of the interchangeability of kinetic parameters in the calculated profiles of the sensor’s signals for various methods of measuring the gas phase (frontal and injection). Thus, according to the results of analyzing the equilibrium gas phase of raw milk samples using an array of sensors with polycomposite coatings, it is possible to predict the quantitative microbiological indicators of raw milk samples with a small error, while the number of sensors can be reduced to six. It is also possible to compare the total profile of the microflora of milk samples from different farms. In future work, it may be possible to increase the specificity of the profiles of the calculated parameters of the sensors with the establishment of the features of the emission of the volatile components of the main pathogenic microorganisms that can occur in milk.

## Figures and Tables

**Figure 1 sensors-24-03634-f001:**
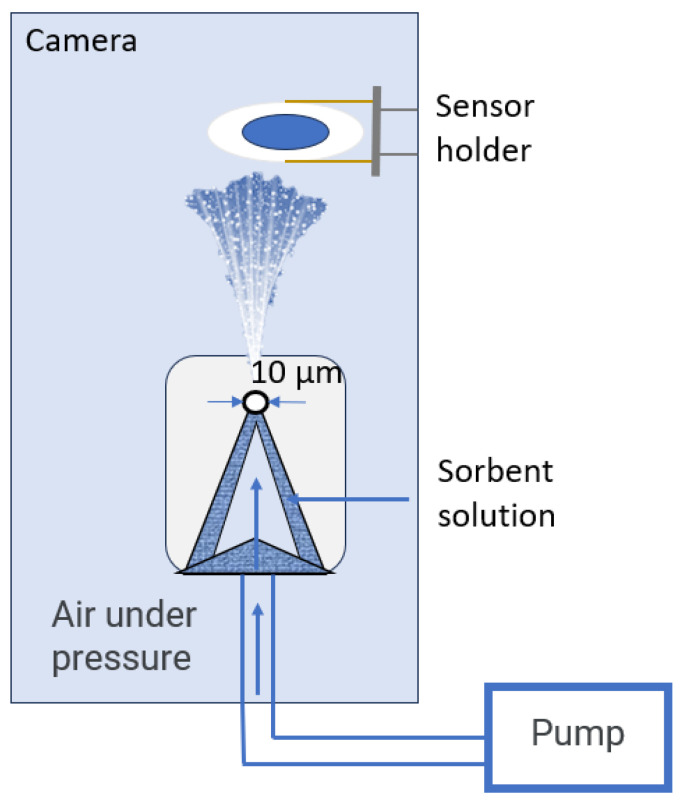
Diagram of the sensor coating plant.

**Figure 2 sensors-24-03634-f002:**
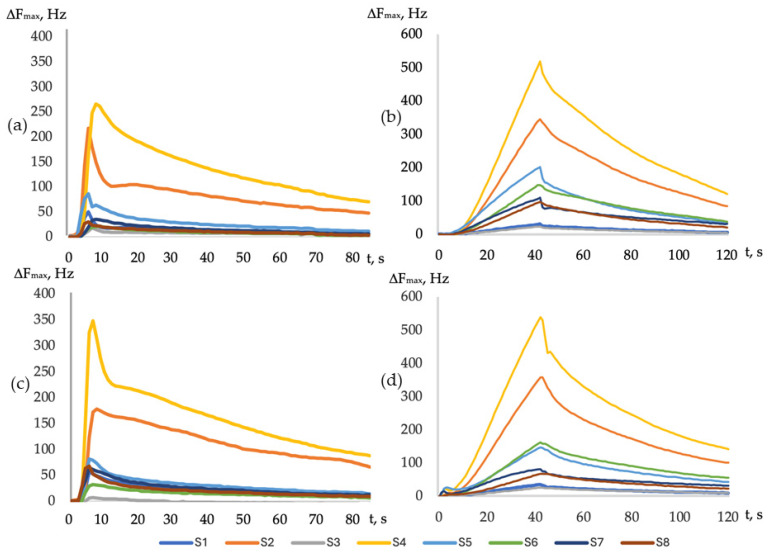
Example of the output curves of milk sample No. 6 (**a**,**b**) and a 0.01% mass aqueous solution of butyric acid (**c**,**d**) for the injected (**a**,**c**) and frontal (**b**,**d**) gas phase input.

**Figure 3 sensors-24-03634-f003:**
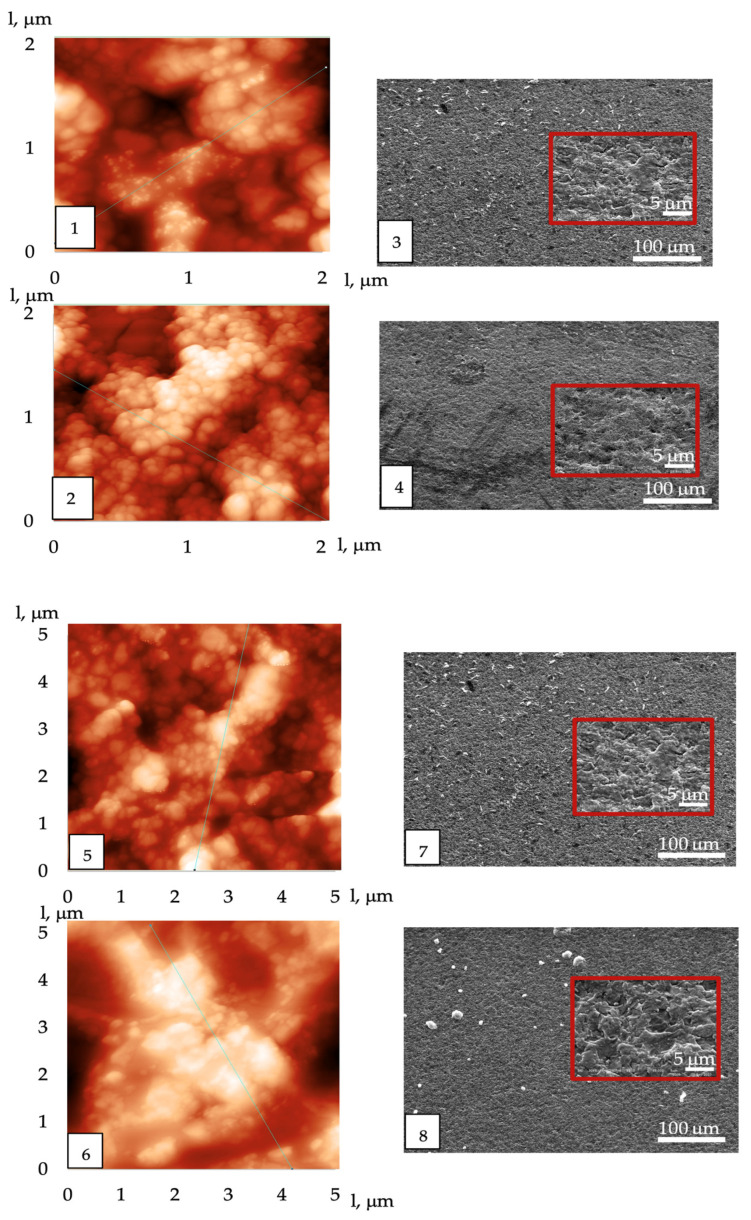
AFM (**1**,**2**,**5**,**6**) and EMS (**3**,**4**,**7**,**8**) photos of the coatings before measurements for choline + sorbitol (**1**,**3**) and choline + erythritol +ASO (**2**,**4**), and after 6 months of operation for choline + sorbitol (**5**,**7**) and choline + erythritol +ASO (**6**,**8**).

**Figure 4 sensors-24-03634-f004:**
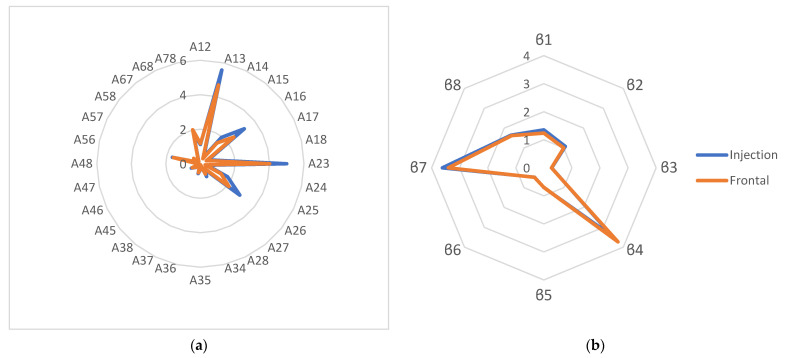
Calculated parameter of the efficiency of sorption Aij (**a**) and the kinetic parameter β (**b**) for milk sample No. 12 with injection and frontal input of the gas phase.

**Figure 5 sensors-24-03634-f005:**
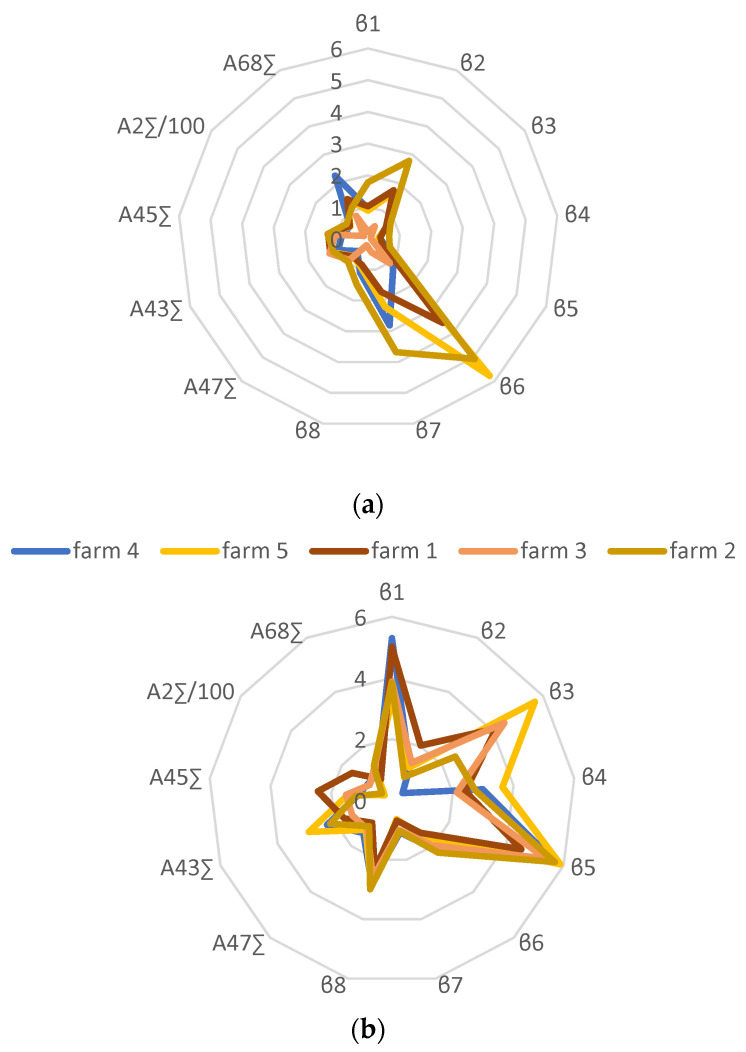
Profiles of the calculated parameters of the sensors for the gas phase of mesophilic anaerobic microorganisms of milk (**a**) and for the gas phase of raw milk samples (**b**) from various farms.

**Table 1 sensors-24-03634-t001:** Characteristics of the sensor coatings.

Sensor Number	Coating	Solvent	Mass, μg	Noise, Hz	Drift, kHz
1	18C6 */chitosan	Toluene	28.7	−2	2.58
2	DHC/chitosan	Ethanol	14.7	−5	1.818
3	Chitosan/CMC	Ethanol	12.5	11	1.41
4	Choline + sorbitol	Ethanol	15.2	2	−0.736
5	Choline + erythritol + ASO	Ethanol	5.29	3	0.519
6	PVP/chitosan	Acetone	12.0	2	1.123
7	PEG2000/chitosan	Acetone	3.41	3	−2.454
8	Erythritol + ASO	Ethanol	7.97	1	0.945

* 18C6, dicyclohexane; DHC, dihydroquercetin; CMC, concentrated micellar casein; PVP, polyvinylpyrrolidone; PEG2000, polyethylene glycol 2000; ASO, amorphous silicon oxide.

**Table 2 sensors-24-03634-t002:** Primer sequences.

Primer Name	Sequence 5′–3′
ITS1	TCC GTA GGT GAA CCT GCG G
ITS4	TCC TCC GCT TAT TGA TAT GC
337F	GAC TCC TAC GGG AGG CWG CAG
1100R	GGG TTG CGC TCG TTG

**Table 3 sensors-24-03634-t003:** Roughness characteristics of the coatings’ based on the AFM analysis.

Covering	Operation Time	Sa *, nm	Ssk	Ska
Choline + sorbitol	0 days	33.46	0.207	0.230
	180 days	71.50	0.712	1.531
Choline + erythritol + ASO	0 days	39.67	0.209	0.155
	180 days	45.25	0.312	0.575

* Average roughness, Sa; surface skewness, Ssk; coefficient of kurtosis, Ska.

**Table 4 sensors-24-03634-t004:** Limits for the determination of volatile compounds (LOD) in aqueous solutions and the specific selectivity coefficient Ks relative to water vapor.

Coating	LOD, mmol/dm^3^				Ks			
	Acetic Acid	Butyric Acid	Isopentanol	Butanone-2	Acetic Acid	Butyric Acid	Isopentanol	Butanone-2
18C6/chitosan	1.8 ± 0.3	1.0 ± 0.4	0.094 ± 0.015	0.56 ± 0.11	1.02	24.9	1.60	1.46
DHC/chitosan	0.17 ± 0.04	10.8 ± 0.5	0.92 ± 0.18	1.11 ± 0.33	1.92	0.24	0.89	0.39
Chitosan/CMC	17.4 ± 3.5	10.6 ± 0.8	9.3 ± 1.6	11.3 ± 3.5	1.44	1.36	0.34	0.07
Choline + sorbitol	2.5± 0.5	1.10 ± 0.07	0.91 ± 0.07	0.12 ± 0.03	1.04	0.94	0.33	0.15
Choline + erythritol + ASO	0.16 ± 0.04	0.11 ± 0.03	0.10 ± 0.02	0.11 ± 0.03	1.02	3.71	1.11	0.49
PVP/chitosan	17.2 ± 3.8	0.12 ± 0.04	0.94 ± 0.06	0.53 ± 0.12	1.13	0.49	0.35	0.12
PEG2000/chitosan	1.16 ± 0.06	0.13 ± 0.05	1.10 ± 0.09	0.11 ± 0.04	1.06	2.07	1.17	0.11
Erythritol + ASO	0.18 ± 0.08	0.12 ± 0.03	0.11 ± 0.02	0.12 ± 0.05	0.87	12.5	1.53	1.42

**Table 5 sensors-24-03634-t005:** Microbiological indicators of the milk samples.

Farm Location	No. Sample	QMAFAnM, CFU/cm^3^	Yeast, CFU/cm^3^	Mold, CFU/cm^3^
Khokholsky district, farm No. 1	1	10.0 × 10^6^	1.0 × 10^4^	0
	4	3.4 × 10^5^	0	0
	14	3.9 × 10^7^	1.0 × 10^4^	0
Khokholsky district, farm No. 2	2	4.0 × 10^6^	1.0 × 10^3^	6.6 × 10^2^
	6	5.9 × 10^5^	6.5 × 10^2^	9.0 × 10^2^
	8	9.8 × 10^7^	8.0 × 10^3^	60
	11	3.5 × 10^7^	1.8 × 10^3^	1.4 × 10^3^
Repyevsky district, farm No. 3	3	4.5 × 10^6^	1.0 × 10^3^	10
	9	4.8 × 10^5^	0	10
	13	3.4 × 10^6^	1.7 × 10^4^	10
Ramonsky district, farm 4	5	2.4 × 10^6^	1.5 × 10^3^	1.6 × 10^2^
	10	5.7 × 10^6^	3.4 × 10^4^	3.0 × 10^2^
Repyevsky district, farm No. 5 Standard No. 2 Standard No. 1	7	4.6 × 10^6^	5.7 × 10^3^	0
	12	2.0 × 10^6^	2.3 × 10^3^	10
	-	1.7 × 10^4^	0	0
	-	6.3 × 10^3^	0	0

**Table 6 sensors-24-03634-t006:** Characteristics of the microorganisms in raw milk samples, identified using PCR analysis.

Farm Location	Type of Microorganism	Identified Species Name	% Coincidence with BLAST Data	Sporulation
Khokholsky district, farm No. 1	Bacterium	*Corynebacterium variabile*	94.56	No
	Yeast	*Clavispora lusitaniae*	95.18	Yes
Khokholsky district, farm No. 2	Bacterium	*Acinetobacter johnsonii*	96.47	Yes
	Bacterium	*Pseudomonas helleri*	93.26	No
Ramonsky district, farm No. 3	Bacterium	*Bacillus thuringiensis*	85.92	Yes
	Bacterium	Uncultured *Acinetobacter* sp.	93.78	No
	Yeast	*[Candida] pseudoglaebosa*	99.83	No
	Yeast	*Clavispora lusitaniae*	97.58	Yes
	Mold	*Geotrichum candidum*	99.7	Yes
Repyevsky district, farm No. 4	Bacterium	*Acinetobacter johnsonii*	97.87	Yes
	Bacterium	*Corynebacterium variabile*	98.61	No
	Yeast	*Trichosporon coremiiforme*	100	Yes
	Yeast	*Rhodotorula dairenensis*	99.65	Yes
	Mold	*Schizophyllum commune*	99.33	No
	Mold	*Aureobasidium melanogenum*	99.27	Yes
Repyevsky district, farm No 5	Bacterium	Uncultured *Acinetobacter* sp.	83.2	No
	Yeast	*Clavispora lusitaniae*	98.03	Yes
	Mold	*Schizophyllum commune*	100	No
	Mold	*Dothiorella gregaria*	98.82	Yes

**Table 7 sensors-24-03634-t007:** Parameters of partial least squares regression for predicting the microbiological parameters of milk samples.

Predicted Indicator	Treatment of Milk	Number of Factors	Optimal Variables	^1^ RMSEP	R^2^	^2^ Δ,%
QMAFAnM	None	3	28, points of the output curves of sensors; 10, calculated parameters	0.390	0.932	4.9
Yeast	Ultrasound	1	12, points; 13, parameters	0.535	0.705	12
Mold	Ultrasound	2	17, points; 15, parameter	0.587	0.721	17

^1^ Root mean square error of prediction; ^2^ relative error of prediction, %.

**Table 8 sensors-24-03634-t008:** Results of predicting microbiological parameters for milk samples from the test set using regression models.

Sample No.	QMAFAnM, lg (CFU/cm^3^)		Yeast, lg (CFU/cm^3^)		Mold, lg (CFU/cm^3^)	
	Predicted	Reference	Predicted	Reference	Predicted	Reference
1	6.70 ± 0.45	6.30	3.70 ± 0.39	3.36	1.71 ± 0.65	1.05
2	6.16 ± 0.27	6.38	3.31 ± 0.31	3.18	2.54 ± 0.98	2.20
3	6.34 ± 0.69	5.77	3.17 ± 0.47	2.81	2.14 ± 0.74	2.95
4	8.05 ± 1.04	7.99	4.08 ± 0.56	3.91	1.36 ± 0.83	1.77

## Data Availability

The data are available from the corresponding author upon request.

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
