# Peer review of "Possibilities of an Electronic Nose on Piezoelectric Sensors with Polycomposite Coatings to Investigate the Microbiological Indicators of Milk"

_sensors, 2024, doi:10.3390/s24113634_

Round 1
Reviewer 1 Report
Comments and Suggestions for Authors
Comments are included in a PDF file.

Author Response
The authors thank the reviewer for their valuable comments, which helped to improve the quality of the article. Explanations of the changes made are below.

Reviewer 2 Report
Comments and Suggestions for Authors
The work presented by Anastasiia Shuba et al., deals with the assessment of microbiological indicators of milk safety, exploiting a piezoelectric sensor array.
The topic could be of interest for a broad readership, and it fits within the journal scope, but the manuscript itself is not easy to follow and the data are not always clearly reported and discussed.
I cannot suggest publication of the manuscript as it is, but substantial revisions are required.
In detail, my comments are as follows:
1) Introduction could be expanded to increase the impact of the work. Authors could focus more on the use of electronic noses in food industry and in particular in milk and diary products, enlightening the state of art and clearly pointing out where their work stands.
Authors should read and cite also:
- Ren, J., Zhou, Y., Zhou, Y., Zhou, C., Li, Z., Lin, Q., & Huang, H. (2015). A piezoelectric microelectrode arrays system for real-time monitoring of bacterial contamination in fresh milk. Food and bioprocess technology, 8, 228-237.
- Sharma, H., & Mutharasan, R. (2013). Rapid and sensitive immunodetection of Listeria monocytogenes in milk using a novel piezoelectric cantilever sensor. Biosensors and Bioelectronics, 45, 158-162.
- Yakubu, H. G., Kovacs, Z., Toth, T., & Bazar, G. (2022). Trends in artificial aroma sensing by means of electronic nose technologies to advance dairy production–a review. Critical Reviews in Food Science and Nutrition, 63(2), 234-248.
Additionally, sentence in lines 56-58 is not completely clear; is punctuation missing after refs [12,14,15]?
2) Around 20% of the bibliography is represented by authors’ articles. Please, lower the number of self-citations, since not all of them seem necessary according to the reviewer. For instance, ref 23 and 24 seem not mandatory at lines 100 and 101, respectively; the same for ref 30 at line 146. Additionally, on behalf of clearness and easiness to follow the work for a reader, the authors could consider to clearly describe in the present manuscript how they assessed sensor noise and drift (line 142), without remanding the reader to ref 29, and how the normalization and assessing of the stability of the array during the measurements have been achieved (line 244), without remanding to ref 40.
3) At lines 148-149 authors wrote “the output curves of the sensors were recorded in the form of chronofrequency, which were analyzed using built-in data processing algorithms (determination of the analytical signal (-ΔFmax,i).”. No curves are shown in the manuscript. The reviewer considers quite important that the authors report in a figure some examples of the output data.
4) Table 3: please point out what the acronyms (i.e. Sa, Skk and Ska) stand for.
5) Sections 2.2.3 and 3: in general, it is quite difficult for a reader to understand the parameters introduced and how or why they are important in the analysis, mainly because there is not a clear explanation of the analysis methods, but the authors often remand to references, as already mentioned. On the behalf of clearness, authors should sum up the main aspect of the analysis and clearly report which parameters they will evaluate and why. In this way they can increase the impact of the paper and allow also readers not completely familiar with such analysis to understand the results.
6) Lines 270-277: Authors wrote “One of the main characteristics of sensor coatings that determine the sorption properties of a sensor is its porosity and surface characteristics.”. Can they better comment and describe in detail how porosity and surface characteristics can influence the sorption properties of the sensors?
7) Figure 2: what are the green lines visible in the AFM images? Additionally, authors should properly comment the obtained morphology in the manuscript. At this time, figure 2 and table 3 are simply reported and not clearly described.
8) Table 4: authors should properly comment the results, supporting the with literature: which is the concentration of the analytes under investigation in a milk sample? Is the limit of detection that they evaluated enough to detect these compounds in the milk sample?
9) Lines 290-293: sentences “Coatings are characterized by low limits for the determination of volatile compounds in aqueous solutions and rather high specific selectivity coefficient to the carboxylic acids. The lowest detection limits for volatile substances are typical for sensors coated with erythritol.” should be supported by references.
10) Line 387 and following: authors wrote: “It is shown that the strongest deviation between samples is typical for sensors No. 3 and No. 6. The parameters of sensor No. 4 are most significant when used in calculations with the parameters of other sensors (No. 5, No. 7, No. 3).”. Can they explain why this occurred?
11) Figure 5 is hard to read; please, consider another way to plot the parameters.
12) Line 395: please, explain the acronyms the first time they appear in the text (i.e. DESs)
13) Lines 397-407: The reported claims are not supported by experimental evidence. Please, provide the data showing (i) the not reproducible signal in the first two weeks; (ii) the reproducibility of the signal after two weeks; and (iii) AFM and SEM images of the samples as prepared and after 6 months.
14) References are not in the same format.
Author Response

(The authors gave the same response as above.)

Round 2
Reviewer 1 Report
Comments and Suggestions for Authors
The text has been revised in accordance with the comments sent to the authors.
Author Response
Thank you for your valuable comments and suggestions.
Reviewer 2 Report
Comments and Suggestions for Authors
A. Shuba et al. properly addressed quite all comments made by the reviewer and they improved the quality of the manuscript and the clarity of the results presentation.
Nevertheless, before publication I still think a couple of points need to be addressed:
1) Figure 2: improve the quality of figure (i.e. same character for numbers on axes and labels…).
2) Figure on AFM: scale bars missing for the AFM images; please add them in all panels. Additionally, it is not clear which panels refer to the images acquired on the samples as prepared or after 6 months. Please, put clear label in the panels and add a clear caption.
3) The Figure shown in the response file to answer the comment n13 needs to be added also in the manuscript or in the SI file. Indeed, it can support the claim made by the authors about the non-reproducibility of the signal in the first two weeks and the reproducibility of the signal after two weeks.
4) the links provided for the externally hosted supplementary information cannot be entered by the reviewer.
Author Response
The authors are grateful to the reviewer for suggestions and comments to improve the paper.
The answers for comments are below:
- Figure 2: improve the quality of figure (i.e. same character for numbers on axes and labels…).
Answer: The figure 2 was improved, all texts in the figure were scaled.
- Figure on AFM: scale bars missing for the AFM images; please add them in all panels. Additionally, it is not clear which panels refer to the images acquired on the samples as prepared or after 6 months. Please, put clear label in the panels and add a clear caption.
Answer: The figure 3 is modified. The panel at each photo is added, and the capture was also modified.
- The Figure shown in the response file to answer the comment n13 needs to be added also in the manuscript or in the SI file. Indeed, it can support the claim made by the authors about the non-reproducibility of the signal in the first two weeks and the reproducibility of the signal after two weeks.
Answer: The figure is added in the text as Figure A1, the information about reproducibility of signal is added into text (lines 394-395).
- the links provided for the externally hosted supplementary information cannot be entered by the reviewer.
Answer: Thank you for the comment, the right link for the supplementary information https://doi.org/10.5281/zenodo.11094735.